# “Ectomosphere”: Insects and Microorganism Interactions

**DOI:** 10.3390/microorganisms11020440

**Published:** 2023-02-09

**Authors:** Ugo Picciotti, Viviane Araujo Dalbon, Aurelio Ciancio, Mariantonietta Colagiero, Giuseppe Cozzi, Luigi De Bellis, Mariella Matilde Finetti-Sialer, Davide Greco, Antonio Ippolito, Nada Lahbib, Antonio Francesco Logrieco, Luis Vicente López-Llorca, Federico Lopez-Moya, Andrea Luvisi, Annamaria Mincuzzi, Juan Pablo Molina-Acevedo, Carlo Pazzani, Marco Scortichini, Maria Scrascia, Domenico Valenzano, Francesca Garganese, Francesco Porcelli

**Affiliations:** 1Dipartimento di Scienze del Suolo, della Pianta e degli Alimenti, University of Bari Aldo Moro, 70126 Bari, Italy; 2Department of Marine Science and Applied Biology, University of Alicante, 03690 Alicante, Spain; 3Faculty of Education and Science, University of Cordoba, Montería 230002, Colombia; 4Institute for Sustainable Plant Protection, National Research Council (CNR), Via G. Amendola 122/D, 70126 Bari, Italy; 5Institute of Food Production Sciences, National Research Council (CNR), Via G. Amendola 122/O, 70126 Bari, Italy; 6Department of Biological and Environmental Sciences and Technologies, University of Salento, 73100 Lecce, Italy; 7Institute of Biosciences and Bioresources, National Research Council (CNR), Via G. Amendola 165/A, 70126 Bari, Italy; 8Faculty of Sciences of Tunis, University of Tunis El-Manar, Tunis 1002, Tunisia; 9Colombian Corporation for Agricultural Research Agrosavia C. I. Turipana-AGROSAVIA, Km. 13, Vía Montería-Cereté 230558, Colombia; 10Dipartimento di Bioscienze, Biotecnologie e Ambiente (DBBA), University of Bari Aldo Moro, 70126 Bari, Italy; 11Research Centre for Olive, Fruit and Citrus Crops, Council for Agricultural Research and Economics (CREA), 00134 Roma, Italy

**Keywords:** alien, invasive or quarantine pest, Integrated Farming, resilience, antifragility, IPM

## Abstract

This study focuses on interacting with insects and their ectosymbiont (*lato sensu*) microorganisms for environmentally safe plant production and protection. Some cases help compare ectosymbiont microorganisms that are insect-borne, -driven, or -spread relevant to endosymbionts’ behaviour. Ectosymbiotic bacteria can interact with insects by allowing them to improve the value of their *pabula.* In addition, some bacteria are essential for creating ecological niches that can host the development of pests. Insect-borne plant pathogens include bacteria, viruses, and fungi. These pathogens interact with their vectors to enhance reciprocal fitness. Knowing vector-phoront interaction could considerably increase chances for outbreak management, notably when sustained by quarantine vector ectosymbiont pathogens, such as the actual *Xylella fastidiosa* Mediterranean invasion episode. Insect pathogenic viruses have a close evolutionary relationship with their hosts, also being highly specific and obligate parasites. Sixteen virus families have been reported to infect insects and may be involved in the biological control of specific pests, including some economic weevils. Insects and fungi are among the most widespread organisms in nature and interact with each other, establishing symbiotic relationships ranging from mutualism to antagonism. The associations can influence the extent to which interacting organisms can exert their effects on plants and the proper management practices. Sustainable pest management also relies on entomopathogenic fungi; research on these species starts from their isolation from insect carcasses, followed by identification using conventional light or electron microscopy techniques. Thanks to the development of omics sciences, it is possible to identify entomopathogenic fungi with evolutionary histories that are less-shared with the target insect and can be proposed as pest antagonists. Many interesting omics can help detect the presence of entomopathogens in different natural matrices, such as soil or plants. The same techniques will help localize ectosymbionts, localization of recesses, or specialized morphological adaptation, greatly supporting the robust interpretation of the symbiont role. The manipulation and modulation of ectosymbionts could be a more promising way to counteract pests and borne pathogens, mitigating the impact of formulates and reducing food insecurity due to the lesser impact of direct damage and diseases. The promise has a preventive intent for more manageable and broader implications for pests, comparing what we can obtain using simpler, less-specific techniques and a less comprehensive approach to Integrated Pest Management (IPM).

## 1. Introduction

Insects have inhabited the Earth for approximately 480 million years, representing the dominant life form as species and biomass [1,2,3,4]. Their presence expresses the high biodiversity of insects in a wide range of ecological niches and results from their genetic plasticity, adaptability, and co-evolutionary processes with other organisms.

The influence between microorganisms and insects has led to the establishment of different types of interaction that can be summarised in various forms of interactions. De Bary [5] first defined symbiosis as “*the cohabitation of distinct organisms*”. Symbiosis is here intended as a significant biological *liaison* between or among species. Symbiosis is one of the leading evolutionary drivers promoting natural biological novelty. Symbiotic relationships between prokaryotes and eukaryotes are present in all kingdoms of life [6].

Symbioses between two organisms can be broadly classified as mutualism, commensalism, or antagonism, depending on the interaction between the species involved. The impact of the symbiont on the amphitryon can highlight an evolutionary continuum between antagonism (negative interaction), commensalism (neutral interaction), and mutualism (beneficial interaction) [7]. Symbiotic relationships drive many interesting biological processes, both within individuals and at the ecological system level.

For the sake of this contribution, we consider ectosymbiont as the guest living out of the host body and endosymbiont as a guest living in the host body. Inside the amphitryon body, the guest may live within specialised host cells [8,9,10].

Mutualism provides an advantage for both species involved. Mutualistic microorganisms can give the host essential nutrients, protection from enemies, increase fitness, and mediate the host’s interaction with other species [11]. Mutualists can be divided into obligate and facultative. Obligate or primary symbionts are microorganisms necessary for the host’s survival. They tend to improve the nutritional aspects of unbalanced diets on which the host feed [12]. Facultative symbionts are not essential for the survival of their host and have broader effects ranging from modifying nutritional aspects to manipulating reproduction. Secondary mutualists can often only be found in a fraction of the host population [11,12].

Commensal symbiosis represents a symbiotic relationship in which one organism benefits without associated costs [12,13].

Finally, parasitism or antagonism represents an unbalanced interaction in favour of the guest microorganism that takes advantage of the insect, generating a loss of fitness or causing host death. Antagonists may be obligate (host-specific) or facultative generalists. Antagonism between insects and entomopathogenic organisms results from co-evolution in which the pathogen aims to host exploitation better and improve its transmission. In contrast, the insect seeks to exclude the pathogen more effectively by improving its defence strategies [14]. Both actors involved in antagonism adopt physiological, ecological, and ethological adaptations to maximise their fitness [15].

Manipulating and modulating these interactions represents an approach for counteracting plant pests and pathogens, mitigating related damages/symptoms that generate considerable food and economic losses. Moreover, the approach reduces the food and feeds contaminating microorganisms, and the toxicological risk [16]. Such interactions also provide a basis to develop future research work in a relatively new field due to the vast diversity of insects and microorganisms in a broad range of trophic and ecological niches worldwide. The chapters below represent distinctive interactions, from the pure endosymbiosis case to the pure ectosymbiotic case. Properly-available techniques make discoveries and rising topics possible, and we consider the methods the real drivers in knowledge evolution [17]. We do not report the endless number of studies treating ectosymbiotic interrelationships, i.e., the studies on microorganisms thriving in the insect gut lumen reviewed by Steinhaus [18]. Moreover, we suggest the 1912 book by Peglion [19] and the 1913 study by Tonelli [20] to be among the first notes on insects transmitting ectosymbiotic plant pathogens.

## 2. Endosymbiotic Bacteria in Pests

Extensive literature on endosymbionts, such as *Wolbachia* spp., *Buchnera* sp., *Rickettsia* spp., *Cardinium* spp., and other species, evidences a close evolutionary link with the insects’ hosts. Known effects include, among others, changes in trophic behaviour and marked effects that manipulate and interfere with the host reproduction and speciation pathways [21,22]. However, few studies have focused on interactions of weevils or *Philaenus* spp. with their endosymbionts. A search on PubMed (https://pubmed.ncbi.nlm.nih.gov/ (accessed on 31 August 2022) with the “*Wolbachia*” query yielded 3740 records, but only 33 by adding the term “weevil”. Similarly, “*Buchnera*” gathered 494 papers, to which including “weevil” in the query added only one. The summary in Table 1 shows the results with different terms and combinations.

No studies are available in PubMed for “*Philaenus*” and “*Bacillus*”, while one is available for “*Cosmopolites*”. Ten were found for “*Rhynchophorus*” and “*Bacillus*”, mainly concerning *Bacillus thuringiensis* Berliner, 1915 antimicrobial activity and host immune response. No studies were found in PubMed when using the term “*Serratia*” with “*Cosmopolites*” or “*Philaenus*”. However, *Serratia marcescens* isolates from *R. ferrugineus* [23,24] exist in publications.

## 3. Pathogens Spread by Monophagous Vectors: *Candidatus* Phytoplasma Vitis

Knowing and studying the biology of a plant pathogen and its vector is necessary for epidemic management, especially when it comes to quarantine pathogens. A quarantine organism is a pest/pathogen of economic importance and yet to be present in an area or present but not widely distributed and officially-controlled [25]. For EU member countries, quarantine organisms fall into “EU relevant quarantine organisms” and “EU priority quarantine organisms”. Once introduced into a territory, “priority quarantine organisms” could have a more severe impact than “relevant quarantine organisms”. Therefore, knowledge of the biology of the pathogen and its vector(s) can provide decisive help in eradication and containment actions. Vector-borne viruses, bacteria, and phytoplasmas are numerous, and many insects can potentially carry these microorganisms. Some hemipterans feed on mesophyll and sap, while others feed exclusively on xylem or phloem sap. This specialisation makes them suitable for transmitting pathogenic pathogens that live in the circulatory system of plants [26].

Vectors may be specific for a microorganism; if the vectors are monophagous, the epidemiological process is straightforward to study and, therefore, hypothetically easier to manage in the field. In some cases, multiple insects may transmit a pathogen. If the vectors are also polyphagous, the epidemiological process is more complex to study and manage effectively [27,28].

An example of a pathogen spread by a monophagous vector is *Candidatus* Phytoplasma vitis, a quarantine organism included in the A2 EPPO list [29]. This organism is the etiological factor of Flavescence Dorée (FD), the most threatening of the Grapevine Yellows (GY) diseases in Europe [29,30].

FD first appeared in the 1950s in south-west France and other areas of Europe, North America, Asia Minor, and Australia [31,32].

The main symptoms of this disease are yellowing, downward curling of leaves, fruit abortion, stunted growth, and lack of lignification of new shoots [33].

The only wild vector of this phytoplasma is *Scaphoideus titanus* Ball, 1932 (Hemiptera, Cicadellidae), which feeds and completes its life cycle exclusively on grapevines [27,34]. *Euscelidius variegatus* (Kirschbaum, 1858) (Hemiptera, Cicadellidae) can only transmit FD under laboratory conditions [35].

*Scaphoideus titanus* is native to North America and entered Europe, presumably by transporting nursery material containing the insect’s eggs [36]. It was first reported in France in 1958 [37] and in Liguria (Italy) in 1964 [38].

*Scaphoideus titanus* is monovoltine and completes its life cycle exclusively on the grapevines. Egg hatching is gradual, beginning in mid-May and continuing until July. Post-embryonic development lasts about 40 days [29,38]. It feeds on the phloem sap of the vine and, during its trophic action on infected plants, acquires the pathogen in addition to nutrients.

Phytoplasmas are obligatory parasites of plants and vectors [29,39], and *Candidatus* Phytoplasma vitis infects the grapevine phloem and various organs of the vector insect (circulatory) and actively multiplies in both hosts (propagative) [26]. *Scaphoideus titanus* can assume phytoplasma both at the nymph and adult instars. In this case, infectivity does not disappear with the metamorphosis, persisting until the insect’s death.

Under laboratory conditions, the adult insect requires a 7-day acquisition access period, a minimum latency of at least 7 days, and a 7-day inoculation access period. When the insect is in the nymphal stage, after ingestion of the pathogen, a latency time of 3–5 weeks is required [40].

After a 13-day capture access period, the capture rate is 91.4% [40]. Vectors are crucial to the outbreak, and studying their feeding behaviour seems pivotal to understanding why some cultivars are more tolerant, as it appears that *S. titanus* prefers to feed on some cultivars over others. This behaviour probably depends on the phloem’s chemical composition, making some cultivars less palatable. Cultivars’ tolerance exists because of the intrinsic ability to deter the vectors [41]. Bressan [42] demonstrated how phytoplasma negatively affects the fitness of *S. titanus*, causing shorter adults lifespan, lower fecundity, and a prolonged egg-hatching time. Endosymbiotic organisms, such as phytoplasmas, are therefore pathogenic for both hosts, i.e., the plant and the insect vector. However, this does not happen for ectosymbiotic organisms, which can develop a disease in plants without compromising any vital aspect of the vector. Ectosymbiotic microorganisms are transported from one plant to another simply by binding externally to the insect’s body without interacting with the internal organs.

Before multiplying in the insect’s organism, phytoplasma must pass the midgut and the salivary glands epithelia. Various glycoconjugates exist on the surface of these tissues, to which many pathogen adhesins bind. *Candidatus* Phytoplasma vitis binds with the VmpA adhesin to N-acetylglucosamine and mannose on the surfaces of the midgut and salivary glands of the vectors. Furthermore, the glycoconjugate patterns are very similar between *S. titanus* and *E. variegatus*, which may partly explain the specificity that *S. titanus* has for *Candidatus* Phytoplasma vitis [43].

With this type of pathogen, given the simplicity of the epidemic process, the management of one or more outbreaks may be easier than others.

EU member countries practice obligatory phytosanitary controls to manage FD spreading with immediate destruction of symptomatic plants and compulsory insecticidal treatments for vector control [44]. Monitoring in northern Italy has shown that the vector is present in large numbers, especially in abandoned vineyards and where there is inadequate pest management [45]. Vector population density is lower in managed vineyards. In Reggio Emilia province (Italy), the estimated vector density was 0.3–0.2 insects per plant in 2008–2009 [46].

Preventive monitoring is ready to eradicate the pathogen and its vector quickly. The control of *S. titanus* is necessary for the success of eradication or containment actions and, considering a reduction in the use of insecticides in agriculture, the in-depth study of pathogen-vector interactions is essential to find new ways of managing an epidemic.

## 4. Genomic Clues in Insect Pathogenic Viruses

Viruses have a close evolutionary relationship with their hosts, being host-specific and obligate parasites. Applying genomic and metagenomic approaches has uncovered several new viruses that remained hidden or have not entered already-described genera or families [47,48]. The research has led thus far to sixteen families of viruses infecting insects. The most studied include *Baculovirus* (Baculoviridae) and *Cytoplasmic Polyhedrosis Virus* (CPV, Spinareoviridae). Other pathogenic viral lineages in insects belong mainly to Reovirinae, Densovirinae, and Entomopoxvirinae [49]. Some viruses are the main ingredients of bioformulations applied for managing and biocontrolling some economically relevant insect pests [49,50]. However, the information available on the biology of insect viral pathogens is only partially exhaustive, given the extent of the phylogenetic radiations of their hosts.

Insect pathogenic viruses are less persistent than chemical pesticides. However, increased awareness of environmentally safe procedures has re-evaluated their use as biopesticides. Compared to synthetic pesticides, viruses offer crucial advantages such as high host specificity, selectivity, and no risk of environmental contamination. Insect pathogenic viruses are large, ubiquitous and manifest high genomic plasticity [51]. The latter property allows them to select for increasing efficacy, persistence, and other valuable characteristics in pest management, including lack of activity *vs* concomitant parasitoids and predators [52].

The analysis of genomes to identify new insect-pathogenic viruses is a relatively recent research endeavour, also driven by the search for novel information on evolutionary processes eventually recorded in sequenced genomes. Genomic data have progressively revealed the natural history of known and new host-pathogen associations, showing increased viral biodiversity—as indicated by the discovery of new species—as well as the introgression, to varying degrees, of viral genetic material into the host genomes. These processes range from the presence of “fossil” genetic fragments of viral origin to the introgression of actively-expressed genes, which in some cases confer a specific advantage to the host, up to the integration of entire genomes [53]. Genetic exchanges between eukaryotes and viruses have often been considered residuals of previous viral infections. In some cases, gene integration processes provide new functions to the host, enriching its specialisation or functional adaptation to new trophic niches or habitats [54]. The production of critical genomic data from pests and the parallel advances of bioinformatics tools make it possible to assess the real impact of exchanged genetic material on host biology and evolution.

The members of the Nudivirus, previously included in Baculoviridae, represent a distinct monophyletic sister group of dsDNA viruses present in several insect hosts [55,56,57]. They have non-retroviral species, such as an endogenous nudivirus integrated into the genome of the brown planthopper *Nilaparvata lugens* (Stål, 1854) (Hemiptera, Delphacidae) [53]. Several nudivirus-like genes exist in different host lineages, including Hemiptera and Hymenoptera, but only one nudivirus pseudogene infects *Philaenus spumarius* L., 1758 (Hemiptera, Aphrophoridae) [58].

Currently, two genomes of *P*. *spumarius* have been made available on NCBI. A search for the amino acid identity of the nudivirus *per os* infective proteins (PIF) in the *P*. *spumarius* genome, performed with TBLASTX [59], showed several positive, albeit short and fragmented, matches, presumably representing possible acquisitions of small genome fragments (Table 2, Figure 1).

Viruses of weevils include the invertebrate iridescent virus 6 (*Chilo* iridescent virus), a single copy, linear dsDNA member of Iridoviridae, which parasites hosts from Coleoptera and other orders. The virus has also experimentally infected *Diaprepes abbreviatus* (L., 1758) (Coleoptera, Curculionidae), a severe weevil pest of *Citrus* spp. in Florida [61]. Other curculionid viruses include two undescribed macula- and bunya-like RNA viruses reported from eucalyptus snout beetles (*Gonipterus* spp.; Coleoptera, Curculionidae) [62], and an *Entomopoxvirus* found in the European spruce bark beetles, *Ips typographus* (L., 1758) (Coleoptera, Curculionidae) and in *Ips amitinus* Wood and Bright, 1992 (Coleoptera, Curculionidae) [63,64,65]. Finally, a severe disease of the red palm weevil *Rhynchophorus ferrugineus* (Olivier, 1791) (Coleoptera, Dryophthoridae) relies on the *Cytoplasmic Polyhedrosis Virus* (CPV), which produces polyhedral inclusion bodies in all host stages, drastically affecting the pest population density levels [66]. Weevils are also vectors of some plant viruses, such as single-stranded RNA *Tymoviridae* [67].

## 5. Interactions between Entomopathogenic Fungi and Pests

Biological control of invasive pests also relies on certain entomopathogenic fungi (EFs) that can infect hosts in agroecosystems and appear suitable for plant protection exploitation. For many years, the search for such species used their isolation from insect carcasses, followed by identification using conventional light or electron microscopy techniques. Thanks to the development of molecular methods, especially DNA sequencing and omics technologies, it is now possible to identify the most crucial EFs species and detect their presence in different ecological niches, including the soil or plant environments.

EFs number around 1000 species [68], the best-known being *Aspergillus* spp., *Penicillium* spp., *Fusarium* spp., and *Acremonium* spp. [69]. Infection usually occurs through propagules that germinate and invade the host body after contact; the invasive mycelium then colonises the host until it dies. Conidiation from emerging hyphae and/or the production of resting propagules follow the host death [70].

Among the EFs primarily used for pest control, some *Beauveria* spp. (Hypocreales, Cordycipitaceae) are widely used against, for example, the coffee berry weevil *Hypothenemus hampei* (Ferrari, 1867) (Coleoptera, Curculionidae) [15], the Asian corn borer *Ostrinia furnacalis* Guenée, 1854 (Lepidoptera, Crambidae), and the sweet potato weevil *Cylas formicarius* (Fabricius, 1798) (Coleoptera, Brentidae) [71]. Many studies have deepened the knowledge about the role of *Beauveria bassiana* (Bals.) Vuill, 1912, as its insecticidal activity is due not only to the hyphae penetrating and spreading in the host body, but also to the effect caused by various toxins [72]. This fungus demonstrated its relevance in banana crops protection from *Cosmopolites sordidus* (Germar, 1824) (Coleoptera, Dryophthoridae) [73,74,75] due to its ability to significantly reduce the weevil survival [76].

*Beauveria bassiana* products are widely applied on banana plantations to manage *C. sordidus* and use pheromone for mass trapping. Another *Beauveria* species, *Beauveria caledonica*, is responsible for the lethal infections of *C. sordidus* in banana plantations in South America. This fungus produces various secondary metabolites and can modulate the pest immune response [76,77]. Studies with *Metarhizium anisopliae* (Metschn.) Sorok, 1883 reported the potential of this fungus in controlling adult weevils [78].

Several studies are underway to control *P. spumarius*, indicated as the main vector of the bacterium *Xylella fastidiosa* Wells, Raju et al., 1986 involved in the OQDS (Olive Quick Decline Syndrome) in the Salento Peninsula (southern Italy). The insect can acquire and inoculate the bacterium from/to different host plants [79]; therefore, it is essential to limit the transmission of *X. fastidiosa* by managing its vector. Recent studies analyse the ability of some *Trichoderma* spp. isolates in decreasing the survival of *P. spumarius* [80]. An innovative IPM approach may include developing EF-based biocontrol actions. EF also represents an essential source of natural molecules capable of affecting *P. spumarius* metabolism and reproduction, thus limiting the pests’ indirect damage to plants [81].

Species of the genus *Trichoderma* are among the most-studied and -used biocontrol agents worldwide. They not only produce benefits as plant growth promoters but also act, with various mechanisms, against other microorganisms in plant defence. Volatile and non-volatile compounds produced by some species of *Trichoderma* can be perceived by the olfactory structures of *P. spumarius* [81], modifying and directing the insect’s food preferences towards other areas of reduced agricultural interest [81,82,83].

Although supported by valid research data, the information available in the literature on the exploitation of EFs as biocontrol agents still needs to be comprehensive. Critical data on the exploitation of EFs as practical means of biological control and information on the mechanisms involved in fungal-insect interactions still need to be included in many world regions. Therefore, efforts are still required to identify and characterise new fungal strains to investigate their entomopathogenic capacity as an alternative to pesticides.

## 6. Multitrophic Interactions of Entomopathogenic Fungi, Crops, and Insects

Insect pathogens were isolated from Mediterranean soils (Alicante, SE Spain) using *Galleria mellonella* L., 1758 (Lepidoptera, Pyralidae) larvae baits [84]. Samples from 61 sites were from agroecosystems and forests, while soils under *Nerium oleander* L., 1753, gave results from natural environments and gardens. Entomopathogenic fungi (EFs) are the most frequent insect pathogens (32.8% soils). *Beauveria bassiana* is the most abundant species (21% soil). *Metarhizium anisopliae* (6.4%) and *Akanthomyces lecanii* (Zimm.) Spatafora, Kepler and Shrestha, 2017 {*Lecanicillium lecanii* (Zimm.) Gams [=*Verticillium lecanii* Zimm.]} (4.8%) are less frequent. *Beauveria bassiana* also scored the highest virulence in a single soil sample (ca. 90% infected insects) and is the most frequent EF (77.8%) in soils under *N. oleander*. Soils from commercial crop fields of food security importance, such as bananas, are also reservoirs of EFs [85]. Reports indicate that *B. bassiana* is a cosmopolitan entomopathogen, especially in warm areas [86]. Economically important pests, such as thrips [87], aphids [88], or pine processionary (*Thaumetopoea pityocampa* [Denis and Schiffermüller, 1775] [Lepidoptera, Notodontidae]) [89], were detected naturally infected with EFs. *Beauveria bassiana* (isolate Bb203) also infected adults of the Red Palm Weevil, *Rhynchophorus ferrugineus* Olivier, 1790 (RPW), in the field (palm groves) just at the first weevil introduction in south-eastern Spain [90]. *Beauveria bassiana* 203 proved more pathogenic to *R. ferrugineus* than strains from other hosts and sources [91]. The strain applied three times at three-month intervals to field palms naturally infested with RPW caused 70–85% insect mortality [92]. Therefore, EFs are present in arid environments and have great potential for IPM of severe insect pests [93,94].

EFs can also colonise plants and plant waste. The latter is the most frequent component of soil organic matter. Evaluation of the growth and multiplication (conidiation) of common entomopathogens rises from inoculation (on almond peels) and gardening (palm waste) substrates obtained from Mediterranean ecosystems by-products of agriculture [95]. The development of entomopathogens depends on the type of substrate. *Akanthomyces lecanii* grows and sporulates well on almond mesocarp, but *Paecilomyces farinosus* (Holmsk.) A.H.S.Br. and G.Sm., 1957 does not. *Beauveria bassiana* uses palm seed nutrients for growth and sporulation, and leaves of the Mediterranean dwarf palm *Chamaerops humilis* L., 1753 promote the growth and sporulation of both *A. lecanii* and *B. bassiana*. The date palm (*Phoenix dactylifera* L., 1753) has a mycobiota that includes-sporulating fungi (*Penicillium* spp. and *Cladosporium* spp.). *Fusarium oxysporum* Schltdl., 1824 saprotroph and an undescribed *Lecanicillium* c.f. *psalliotae* (Treschew) Zare and W. Gams, 2001 entomopathogen colonise leaves infested with Marlatt red-scale (*Phoenicococcus marlatti* Cockerell, 1899—Hemiptera, Phoenicococcidae) [96]. Palm pathogens, entomopathogenic and saprotrophic fungi strongly interact with each other; *B. bassiana* strongly inhibits *Penicillium vermoesenii* [=*Nalanthamala vermoesenii* (Biourge) Schroers, 2005] (Figure 2), a fungal necrotrophy of palms.

EFs (*B. bassiana*, *Lecanicillium dimorphum* (J.D.Chen) Zare and W.Gams, 2001, and *Lecanicillium* c.f. *psalliotae*) artificially inoculated in living plants act as true endophytes [97]; fungi survive and spread in date palm (*P. dactylifera*) petiole tissues (parenchyma and vascular tissue) at least 30 days after inoculation. *Beauveria bassiana* is a natural endophyte from date palm roots [98]. This fungus was isolated from the roots of date palms in two coastal dune sites with high and low human impact in south-eastern Spain. Root colonisation by endophytic insect-pathogenic fungi has recently been reviewed [99]. Root and microbiota respiration [100] depletes oxygen in the rhizosphere. Fungal parasites of invertebrates, such as the nematophagous *Pochonia chlamydosporia* (Goddard) Zare and W. Gams, 2001 or the entomopathogens *B. bassiana* and *M. anisopliae*, breach chitin-rich barriers to infect the host. These biocontrol fungi can also ferment chitosan, a chitin derivative [101]. Apart from their application in biofuel production, this trait can be an adaptation for survival and insect infection by EFs in the rhizosphere. Entomopathogenic fungi are part of phylloplane and rhizosphere mycobiomes. Their endophytic behaviour allows them to colonise plant-derived substrates, affecting plant-volatile emissions during insect infestations [102]. Plant-derived substrates, such as rice grains, can be used for mass production and formulation of EFs [103,104].

Based on previous reports (see above) on the endophytic behaviour of EFs, several studies tested the response of palms to inoculation with these biocontrol fungi. *Beauveria bassiana*, *L. dimorphum*, and *L.* cf. *psalliotae* induced proteins in plant defence or stress response [105]. The plant immune system responds to microbe-associated molecular patterns (MAMPs) derived from conserved structures (i.e., cell walls) of plant pathogens such as chitin [106]. Chitosan can permeabilise the membrane and kill plant pathogens such as bacteria and fungi in its deacetylated form [107]. EFs and nematophagous fungi (NFs) are compatible with chitosan since they have evolved low-fluidity membranes [108,109] and branched cell walls rich in β-1,3-glucan [110]. Moreover, EFs and NFs are in contact with chitin during host (insects and nematodes, respectively) infection. Chitosan modifies the transcriptome and biology of fungi and plants, causing cell stress [111]. Chitosan can enhance the pathogenicity of fungal parasites of nematode eggs [112,113,114]. These are close relatives of EFs, such as *Metarhizium* spp. [115]. Tests will explain the effect of chitosan on the EFs’ pathogenicity.

Acoustics reveals that RPW larvae have briefer movement and feeding activity with *B. bassiana* infection [116]. We also have evidence that *B. bassiana* formulates used for RPW biocontrol in the field [92] repel adults of this insect pest [117]. Evidence suggested investigating entomopathogenic fungi and close fungal pathogens of invertebrates for volatiles capable of modifying the behaviour of insects of economic importance, such as weevils. Entomopathogenic fungi and close relative nematophagous fungi (*Pochonia* spp. egg parasites) emit volatile organic compounds (VOCs) capable of repelling *C. sordidus* [85] and RPW [117]. P201930831 and P202230103 insect repellents patented VOCs are on field trial for efficacy.

Finally, EFs are a component of plant and soil microbiomes. They are efficient insect pathogens with a multitrophic lifestyle, including plant endophytism, inducing plant defences and modifying insect pest behaviour with their VOCs, which work as low environmental impact tools for insect pest management.

## 7. Native Entomopathogenic Fungi Used for Microbial Control of the *Rhynchophorus palmarum* (L., 1758)

In South America, economic palms such as the coconut (*Cocos nucifera* L., 1753) and the oil palm (*Elaeis guineensis* Jacq., 1897) are crops with social significance for the region. Industrial exploitation offers various raw materials for the cosmetics and food industry, in the settlement as construction materials, and in the traditional use of fresh coconut. Industrial processing induces employment and income opportunities for the community [118,119].

The incidence and damage of insect pests and plant diseases, which causes recurring losses on the farm and lower productivity [120,121], limit the overall production. The South American Palm Weevil (SAPW), *Rhynchophorus palmarum* (L., 1764) (Coleoptera, Dryophthoridae), causes crucial economic losses due to the cryptic larvae that burrow tunnels within the central cylinder of the palm stipes and apical meristem. The SAPW is black and sometimes reddish because of atypical colour polymorphism. It measures between 35 and 60 mm, presents sexual dimorphism between male and female, and the male snout is straight and robust. The male has stout brush-like setae on the front-clypeal head region, while the female rostrum is slender, lacking setae, and slightly arched dorso-ventrally [122,123]. A second species, *Dynamis borassi* (Fabricius, 1802) (Coleoptera, Dryophthoridae), is similar enough to avoid leading any experts to misidentification. The presence and collection of adult specimens of *D. borassi* on Amazonian palm species, *Astrocaryum carnosum* F. Kahn and B. Millán, 1992, and *Astrocaryum chonta* Mart., 1844, provide information on weevil biology obtained from pupal cells collected in damaged inflorescences. The larvae were parasites by *Billaea rhynchophorae* (Blanchard, 1937) (Diptera, Tachinidae), which emerged from the pupal cells [124].

The geographic distribution of SAPW encompasses the Americas, from Argentina to California, and includes the Central American Antilles [125]. SAPW affects the primary area of commercial palm production on the continent and Brazilian regions of economic coconut and oil plantations [121]. SAPW spreads the nematode *Bursaphelencus cocophilus* (Cobb) Baujard (Rhabditia, Parasitaphelenchidae), which is responsible for inducing Red-Ring Disease (RRD) in palms [126,127]. The symptoms of RRD in palms are reddish lesions that gradually form in the stem [121].

The management of SAPW and RRD in coconut and oil palms is complex. However, chemical control has low efficiency in disrupting the SAPW-RRD association. An attempt at agronomical control uproots and burns the affected trees and reduces the infestation. However, this is a post-damage control action with a relevant significant environmental impact that also consists of greenhouse gas production. A more effective control action consists of the mass adult trapping by rhinchophorol coupled to traps with Synergic Blends of Attractive Sources (SBAS) and removing RRD-infected palms by keeping the RRD at low levels [122,128,129,130,131].

Concern over the mass trapping and felling of palms also suggests *in situ* biocoenosis studies identify new or neglected entomopathogenic microorganisms [132,133]. Highly virulent species and strains of native EFs can serve as effective bioinsecticides. Fungal strains native to the environment where they will be applied are fungi that have co-evolved with their host insects, such as certain strains of *B. bassiana* and *M. anisopliae*. These two represent the most widely-used entomopathogenic fungi in biological control [134,135,136]. Significant genetic diversity exists among the available collections, with a wide range of hosts and relevance to tropical and subtropical environments [137].

EFs play a central role in the Brazilian biopesticide market; these fungi mainly work in management of sugarcane spittlebugs or whiteflies in row crops via registered microbial formulation of *M. anisopliae* and *B. bassiana* [138]. That the number of registrations of biological formulations for pest control in Brazil is increasing (Table 3) [139] suggests the collection of relevant details among native biocontrol candidates.

Recent research to control SAPW in Brazil has identified several native strains of highly virulent *B. bassiana* that can be differentiated to minimise resistance [140]. The criteria for selecting isolates for biocontrol originate in the insect mortality rates observed in bioassays and the efficiency of conidia production in the culture medium [141]. Several techniques allow fungi identification; the alpha taxonomy facilitates the clustering of collections, thus enabling the estimation of performance during pathogenicity tests [142] and subsequent molecular studies that are of great importance to identifying and characterising a single native EF strain.

Advancements in molecular techniques, especially those based on DNA analysis by PCR, have enabled the development of rapid, accurate, and applicable methodologies for examining large samples to detect and identify different entities [143,144]. DNA profiles are powerful and sensitive tools to identify fungal isolates infecting a target population [143]. Sequencing ITS (Internal Transcribed Spacer) specific region is a routine technique to understand the phylogeny of EFs. Nuclear markers have highly conserved sequences and serve as barcode regions for identifying fungal species. Sometimes features have low resolving power, e.g., in some groups of ascomycetes [144,145]. The use of different loci, such as α-TEF (Translation Elongation Factor-1α), the nuclear intergenic region of the B locus (Bloc), and the larger subunits (RPB1 and RPB2) of RNA polymerase II, among others, helps [144,145,146].

The EF species with the most potential for development as bioinsecticides are those cosmopolitan ones in the environment where the microorganism will be applied [133]. Exotic species of EFs used in biocontrol may be ineffective in some pests due to adaptation to climatic diversity and differences in isolates from the host. Identifying native EFs is a promising alternative, especially concerning ecological suitability with native pest species and the more negligible effect on non-target organisms than exotic isolates [133,138,146].

## 8. *Bactrocera oleae*, *Colletotrichum* spp. Ectosymbionts and Olive Anthracnose in Mediterranean Areas

Olive (*Olea europaea* L., 1753) suffers from abiotic adversities, pest infestations, and bacterial and fungal or virus infections, hosting many non-pathogenic microorganisms [147,148,149]. The olive fly, *Bactrocera oleae* (Rossi, 1790) (BO; Diptera, Tephritidae; former *Dacus oleae*), is a key pest of olive groves in the Mediterranean basin [150]. This pest thrives where cultivated trees grow extensively, and wild trees are indigenous [151]. We presume that agriculture was a significant driver for the expansion of cultivated and wild olive trees as sources of food, wood, and cattle fodder, despite the relationships between cultivated and wild olive trees in the Mediterranean still being determined.

The literature suggests that the interactions of the olive fly with fungal pathogens belonging to the genus *Colletotrichum* can have a significant economic impact on production [152]. *Colletotrichum* spp. are causal agents of Olive Anthracnose (OA). The species complexes *Colletotrichum boninense* Moriwaki, Toy.Sato and Tsukib., 2003 and *Colletotrichum gloeosporioides* (Penz.) Penz. and Sacc., 1884 can induce OA and impact orchard production in terms of quality and quantity [153]. *Colletotrichum* spp. are considered the most devastating fungal disease of olive trees, more aggressive in areas or conditions of high relative humidity [152]. Moreover, *Fusarium* spp. and *Alternaria* spp., together with some species of the Botryosphaeriaceae, can participate in drupe rots.

Fungal vectors [154], drivers [155], or spreaders could promote the fungal infection of drupes during ripening. Furthermore, oviposition wounds [156] may facilitate the fungal infection process, as in the case of the olive fly. However, injuries are not essential for infection of *Colletotrichum* spp. [157]. Koronéos [158] confirmed the indirect responsibility of the *B. oleae* in opening the way for the *Camarosporium dalmaticum* (Thüm.) Zachos and Tzav. Klon., 1979 (= *Sphaeropsis dalmatica*) in olives via the oviposition wounds. Koronéos also confirmed that the *C. dalmaticum* and the *Lasioptera berlesiana* (Paoli, 1907) (Diptera, Cecidomyiidae) are almost always present.

Interactions between insects and fungi participate in the ecological context, crop production, and human health [159].

Climate change and global warming expand the olive fly [160] northern limit in most countries. However, the Lake Como area remains favourable because of mitigated winter temperatures. Northern Italy and the Apennines stay unfavourable due to winter cold weather dropping below 0 °C. Climate change erodes the olive flies’ territories in southern regions due to lethally high summer temperatures [160].

*Colletotrichum* spp. may enter the drupes directly through the epicarp, but the severity of symptom expression and infection rate may increase if the BO injures the drupes [161]. In many European olive-growing areas, a correlation between the incidence and severity of infections and *B. oleae* infestation is observed, probably due to the action of insect vectors or other spreaders of *Colletotrichum* spp. conidia.

The larval activity of *B. oleae* favours the infective process of *Colletotrichum* spp. and causes early fruit ripening, while the insect contributes to the spread of conidia [161].

*Bactrocera oleae* is also associated with the bacteria [162,163]. Adults and larvae host a non-cultivable bacterium (*Candidatus* Erwinia dacicola), considered an obligate symbiont of the BO [163,164]. However, other bacteria are usually found in the digestive tract of wild olive flies and are probably transient residents ingested with the diet [165,166].

However, despite some direct evidence demonstrating the contribution of bacteria to larval development [167,168], the bacteria’s roles in the BO’s nutritional ecology still need to be resolved.

In general, interactions between organisms have an impact on their evolutionary history. In eukaryotes, insects and fungi predominate in abundance and species diversity [169]. Well-known cases of associations between insects and fungi occur in different ways, such as in the case of bark and ambrosia beetles [170], ants and termites cultivating fungi [171], or yeasts found in the gut of insects, wood wasps, and gall midges [172]. Spores or mycelia that insects ingest or mechanically carry can reach uninfected plants [173].

In plants, insects are involved in disease development through different types of action. Some of these are as follows. (I) Insects visit plant-infected organs exuding bacteria [174] of fungal conidia which dirty the arthropod bodies that spread them to other plants. (II) Insects can wound fruits, leaves, branches, shoots, stems, and roots, opening pathways for pathogens while feeding or laying eggs [175]. (III) Insects weaken plants by probing on them [176] and make plants more vulnerable to pathogens. (IV) Insects are also true vectors acquiring and transmitting propagules of fungi [177], bacteria [178], viruses [179], phytoplasmas [180], and protozoa [181]. Insect vectors transport pathogens from infected to uncontaminated plants by a well-defined and deterministic chain of events, initiating a new infective process and disease. The insects’ active dispersion ability to find the appropriate host increases the effect of pathogens spreading [182]. However, the speed and mode of dispersal of pathogens, and their role in epidemics, depend on the type of contamination mechanism of the insect’s body, either external (mechanical vectors) or internal (biological vectors) [183,184].

The role of the weevil *C. sordidus* in the epidemiology of the *Fusarium* sp. wilt of bananas in the field remains uncertain. Meldrum [185] considered the weevil an external spreader, but we need data on the possible presence inside the weevil body. Furthermore, the pathogen’s dynamic of acquisition and permanence remains unknown, while it is a crucial factor in the role of *C. sordidus* in pathogen dispersion [186].

Moreover, introducing new pathogens is only necessarily followed by disease emergence if a second factor spreads the pathogen. Without specific vectors, some pathogens may remain localised and cause no disease once introduced to new areas.

Many intra- or extracellular ecto- or endosymbionts thrive with insects [187]. The ectosymbiotic microbiome is being studied in the insect model *Bactrocera*/*Colletotrichum* to mitigate indirect damages [188]. Through molecular approaches, such as ITS and 18S rRNA sequencing, the fungal community of the insect is investigated, although their role still needs to be fully understood [188]. For instance, an analysis of the ITS base of the fungal gut microbiota of BO allowed the identification of a core formed by sooty fungi (*Cladosporium* spp.), plant pathogenic fungi (*Colletotrichum* spp.), and other less abundant Taxa [189,190].

The Metagenomic data are scarce for the bacteriome, mycobiome, and virome of pests such as *Bactrocera* and beneficial predators [191]. Few data are available on host switching rates and multitrophic interactions involving ectosymbionts.

Moreover, the location in the insect body and the type of transmission of associated microorganisms during the vector life cycle or among individuals in the population requires a model study. We consider ectosymbionts those species regularly associated with the host insect occupying specialised insect structures or modifying the insect cell, tissue, organ, or system. Ectosymbionts could show morphological or behavioural changes needed to interact with the insect. Ectosymbionts of insects require recognising opportunistic, occasional, and symbiotic hosts to proliferate on the external cuticular surface.

Bubici [192] studied the microbiome profile of *Aleurocanthus spiniferus* (Quaintance, 1903) (Hemiptera, Aleyrodidae) on *Ailanthus altissima* (Mill.) Swingle, 1916. *Aleurochantus spiniferus* has a wide host range, but the shift to a new host, *A. altissima*, could be associated with new endosymbionts. Specific methods based on morphology and molecular approaches will help to identify ectosymbionts, i.e., studying insect-associated microorganisms by light microscopy, SEM, Cryo-SEM, and combined with Next-Generation Sequencing (NGS).

MinION (Oxford Nanopore Sequencing Technology) characterises microbiome samples [193], identifies each microbe, and generates complete and closed genome assemblies, thus elucidating gene expression within microbial communities. Long Nanopore sequencing reads will provide improved genome assemblies, accurate identification of closely-related species, and detailed analysis of full-length RNA transcripts from mixed microbial samples. Data will be provided in real time, allowing immediate access to species identification, abundance, and antimicrobial resistance results.

Advances in mitigating damages and disease incidence would include the study of the pest itself and the characterisation of its ectosymbionts. Interfering with the microorganism’s life cycle offers a new strategy for sustainable and effective management practices.

Moreover, the study of the gut microbiota of insects is of great interest in medical research and economic exploitation in agricultural production [194]. Therefore, studies on the gut microbiota elucidate the identification of new ways to control crop pests, demonstrating that changes in phylogeny or diet can modulate insect populations and influence host fitness.

The gut microbiota plays a nutritional role in pests, as olive fly larvae depend on their gut microbiome to break down the phenolic compounds in unripe fruits [195].

Evaluating the effects of endo- or ectosymbionts on their hosts and consequently on plants and pathogens by studying insect-microbiota will contribute to a better understanding of insect ecology and explain their success in nature.

## 9. Grape Berry Moth (*Lobesia botrana*) Interaction with *Botrytis cinerea* and Ochratoxigenic *Aspergillus* Species

The European Grapevine Moth (EGVM) *Lobesia botrana* (Denis and Shiffermüller, 1775) (Lepidoptera, Tortricidae) is one of the main pests affecting vineyards worldwide, including all the European grape-growing areas [196]. This Lepidoptera is also present in America, where it is a quarantine pest subject to official control [197,198]. EGVM performs 2–4 generations per year on *Vitis vinifera*, depending on latitude, climate, and microclimate. The first generation of *L. botrana* is anthophagous, and developing on floral clusters is of lesser economic concern [199]. In contrast, the subsequent carpophagous generations thrive on berries during the early ripening process. The direct damage of the carpophagous larvae is caused by larval feeding on the unripe grape berries, resulting in a loss of grape weight and an unmarketable crop. However, the most significant damage originates from fungal and bacterial infections that drastically reduce the quality [196,197,198,199,200]. Penetration holes on ripe grape berries by third-generation *Lobesia botrana* larvae promote the occurrence of several fungal and bacterial rots: we concentrate on fungal rots. Fungal rots can originate from infections by *Alternaria* spp., *Cladosporium* spp., *Penicillium* spp., *Rhizopus* spp., and grey mould caused by *Botrytis cinerea* Persoon, 1801 and *Aspergillus* black rot caused by black aspergilli species of Section *Nigri* [201,202,203]. The association of EGVM with grape grey mould led to one of the most important grape syndromes in the world and caused severe losses in table grape yields, provoking bad flavours and ruining bouquets in wine. The interaction of the *L. botrana* larvae with the *Aspergillus* black rot on clusters is also considered the primary source of ochratoxin A (OTA) in grapes [204,205]. OTA is the only mycotoxin for which a maximum regulatory level has been established in wine [206], making *Aspergillus* highly detrimental to viticulture.

On grape bunches, the larvae of *L. botrana* often associate with grey mould. Caterpillars can contribute to spore dispersal or act as spore drivers by trapping conidia in the cuticle ornamentation and faeces [201,202]. In addition, larvae-feeding wounds on grape berries promote the rapid colonisation of *B. cinerea*. For these reasons, larval activity spreads the grey mould under field conditions [207,208]. Research has shown that the presence of grey mould on grape berries increases the aggressivity and fitness of larvae and females [209,210,211]. According to Mondy et al. [209,211], *B. cinerea* is attractive to adults and first-instar larvae. The mould promotes the EGVM populations by increasing the survival and fecundity of larvae and reducing their development time. However, other authors have not observed any of these positive effects of grey mould on *L. botrana* populations [212,213,214].

*Lobesia botrana* is a significant risk factor for OTA under field conditions in a vineyard [215,216,217]. OTA is a secondary fungal metabolite, such as alternariol, alternariol monomethyl ether and tenuazonic acid [218,219]. OTA is nephrotoxic and hepatotoxic, in addition to other toxic properties. It is classified as potentially carcinogenic to humans (Group 2B) by the International Agency for Research on Cancer [220]. *Aspergillus* bunch rot is a fungal disease that affects pre-harvest grapes. A complex of *Aspergillus* species in section *Nigri*, including *A. carbonarius*, *A. niger* and *A. tubingensis* [221,222], cause the bunch rot. The importance of these black aspergilli rose when OTA was found as a contaminant in grapes and grape-derived products [223,224]. These ochratoxigenic fungi are opportunists (saprophytes) that cause effects that are not always visible and commonly linked to limited yield losses [225]. Although they are always present in the field, they may develop massively on berries damaged by abiotic and/or biotic causes, from veraison to harvest, with a high incidence at ripening. This contamination is strongly related to climatic conditions, geographical regions (the southern Mediterranean climate is very favourable), vines cultivars and damaging pests [217]. Therefore, significant variations may occur from one year to the next. Where *L. botrana* completes three generations per year and climatic conditions favour the infestation of EGVM larvae from the early veraison to the ripening stage, the control of the third generation is a crucial factor in reducing bunch damage. It also reduces rot at harvest [205,206,207,208].

Currently, applying an effective, economical, and eco-friendly technique to control these agents simultaneously is impossible—most of the control strategies used so far rely on chemicals. However, the use of pesticides is increasingly discouraged due to the environmental pollution problems associated with high application rates. Pesticides could reduce biodiversity, the potential loss of key species such as bees and biological control agents, and even the generation of resistance in some invertebrate pest species [226]. Entomopathogenic fungi could represent an alternative solution for controlling these agents. These organisms are crucial natural control agents that limit insect populations in many natural and artificial ecosystems [227]. Many entomopathogenic fungi infect eggs, immature instars, and adults of many insect species [228]. Studies have evaluated the efficacy of entomopathogenic fungi strains of genera *Beauveria* and *Metarhizium* on *L. botrana* [205,229,230] and the antagonistic effect of *Metarhizium anisopliae* on *B. cinerea* [230].

Further surveys will find new entomopathogenic fungi candidates for use in the biological control of *L. botrana*. New candidates should also consider their antagonistic activity towards *B. cinerea* and black aspergilli. Moreover, their compatibility with fungicides commonly used for grapevine diseases should be assessed.

## 10. Pathogens Spread by Polyphagous Vectors: *Xylella fastidiosa*

Another almost opposite example is the *X. fastidiosa* (XF) epidemic. This bacterium has been known for many years in North America, where it was first isolated from grapevines affected by Pierce’s disease [231]. It is the cause of numerous high-impact diseases and can infect over 550 plant species (80 families), although most remain asymptomatic [232]. *Xylella fastidiosa* is a quarantine pathogen registered in the EPPO A2 list. Its dangerousness is due to the enormous economic and landscape damage it can cause to an area. A strain of this bacterium, the one that causes the citrus disease defined as Citrus Variegated Chlorosis (CVC), is even included in the list of biological agents regulated by the US Agricultural Bioterrorism Protection Act of 2002 [233].

*Xylella fastidiosa’s* first European issue was in 2013. Olive trees showing severe symptoms of desiccation [234] appeared in the province of Lecce (Italy). After the first European report, XF was also found in other EU and non-EU countries [235]. *Xylella fastidiosa* has three subspecies, each with a proper host range. The three main subspecies are subsp. *fastidiosa*, subsp. *multiplex* and subsp. *pauca* [236].

The transmission of the pathogen occurs with the help of xylem sap feeders. Insect vectors of XF belong to the suborder Cicadomorpha, and ca. 50 species have been identified worldwide [28]. The insect species transmitting XF are polyphagous on herbaceous and arboreal plants. They spend the juvenile instars feeding on herbaceous host plants. When they become adults, they also move to bushes or tree hosts [237,238].

Adult vectors that feed on xylem sap from an infected plant acquire the bacterium. Subsequently, the bacterial cells multiply, forming microfilm in the foregut vector lumina [182,239,240]. A non-circulative interaction exists [26] between the bacterium and the adult, where the XF behaves like a non-mutualistic ectosymbiont [182]. *Xylella* is peculiar among plant pathogens because there is no latency period after the acquisition. It has a propagative behaviour and a persistent presence in the adult foregut [241]. In the interaction between the bacterium and the vector, XF is the only one to benefit from. How vectors benefit from the association with *Xylella* is not clear, and the presence of bacterial cells in the precibarium changes feeding behaviours [242].

*Xylella fastidiosa* has many vectors, but some are more important because they are more widespread and efficient in their context. For the United States, *Graphocephala atropunctata* (Signoret, 1854) (Hemiptera, Cicadellidae) is the primary vector in the coastal areas of California, known mainly for spreading Pierce’s disease [243]. *Draeculacephala minerva* Ball, 1927 (Hemiptera, Cicadellidae) is known for the spread of Almond Leaf Scorch (ALS) disease in central California [244]. In North America, the main *Xylella*-vector is *Homalodisca vitripennis* (Germar, 1821; Hemiptera, Cicadellidae) [245]. *Homalodisca vitripennis* is native to the southern USA and northern Mexico and has spread throughout the Americas through the displacement of plant material hosting its eggs [246]. Despite its low transmission efficiency, its extreme polyphagia and ability to travel great distances make it one of the most critical and dangerous vectors of *X. fastidiosa* [247]. *Homalodisca vitripennis* is also present in Oceania. *Homalodisca vitripennis* has yet to be detected in Europe and is included in the EPPO A1 list [248]. In Brazilian *Citrus* spp. groves, the main *Xylella*-vectors are *Aonidiella citrina* (Coquillett, 1891) (Hemiptera, Diaspididae), *Dilobopterus costalimai* Young Jr., 1977, and *Oncometopia facialis* (Signoret, 1854; Hemiptera, Cicadellidae) [249]. The primary vector in Europe is *P. spumarius*, a ubiquitous insect that effectively transmits *Xylella*, capable of rapidly spreading the bacterium [250]. It is considered the main cause of Salento’s Olive Quick Decline Syndrome (OQDS) impact [233].

In the past, researchers thought that transmission of XF occurred without any specificity between vector and pathogen. However, recent studies demonstrate the implication of cell-to-cell signals in XF to colonise insect vectors’ foregut [251,252]. Through the rpfF gene, XF regulates the production of small signal molecules called DSF (Diffusible Signal Factor), which depend on cell density [242]. When these molecules produced by individual bacteria accumulate in an environment, they cause a change in rpfF gene expression, stopping DSF production. When DSF production is blocked, the bacterium cannot effectively colonise the precibarium [242].

Bacterial adhesins and foregut surface carbohydrates play a role in vector-pathogen interactions because affinities depend on the polysaccharides. For example, N-acetylglucosamine inhibits cell adhesion to the chitin substrate [253]. Molecules influence the initial attachment of bacterial cells on the surface of the vector’s foregut. The hemagglutinin-like proteins are decisive for XF adhesion to vector foregut polysaccharides [253]. In addition to haemagglutinins, XF produces other fimbrial and non-fimbrial adhesins. HXFA and HXFB appear essential for the first adhesion and colonisation of the foregut. FimA is involved in adhesion and aggregation [254].

Therefore, epidemics of XF are often very complex phenomena governed by many factors, such as the host plant species and vectors present and their context. Given the threatening nature of XF, in countries at risk of introduction, it is necessary to implement controls on imports of plant material, continuous monitoring, and in-depth studies on the presence of xylem-feeding leafhoppers, known as *Xylella*-vectors. Studying the relationships between *Xylella* and its vectors and how to use this knowledge to develop plant protection products or epidemic management techniques is also necessary.

Furthermore, severe epidemics caused by pathogens such as XF have a considerable social impact. Therefore, XF requires constant education about quarantining plant pathogens and their impact on the territory. In the absence of proper political-scientific communication, binding control actions aimed at eradication or containment of the pathogen could be slowed down by the opposition of citizens and farmers, thus favouring the progress of the epidemic [233].

## 11. Detection of *Xylella fastidiosa* subsp. *pauca* from the Insect Vector

Attempting to prevent the further spread of *X. fastidiosa* subsp. *pauca* ST53 in Apulia, the NPPO provides surveys in the “containment” and “buffer” areas, allowing olive trees sampling by appropriate spoiling techniques on the symptomatic olive foliage. Furthermore, Real-Time PCR with specific primers and following the EPPO procedures for detecting *X. fastidiosa* [255] allowed the detection of new infection foci. However, this method reveals the presence of the pathogen at a time that does not reflect the primary inoculation carried out by the insect vector through its feeding activity on the olive. Indeed, the inoculation of the pathogen could have occurred several months before the inspector collected the samples. Afterwards, one or more infected vectors may have reached other olive trees a few meters away, transmitting XF. This risk seems to increase if the sampling of olive leaves occurs in the year following the insect’s feeding activity [256]. Therefore, this type of survey does not allow XF to be intercepted directly from the insect body, leading to an underestimation of the precise limit of the infection, allowing the bacterium to spread further in the territory. To reduce data uncertainty and better track the bacterium spreading in olive orchards, we suggest including *P. spumarius* adults sampling in olive groves from natural environments.

Scrutiny for the presence of the pathogen by the same EPPO procedures should run during spring (i.e., from the end of April) and autumn (i.e., September and October).

## 12. Bacteria Symbionts of Red Palm Weevil

Studies on insect-microorganism iteration are steadily increasing, especially those focusing on the role of bacteria (as obligate or facultative symbionts) in the life cycle of their hosts [257,258,259,260]. Indeed, bacteria symbionts provide essential nutrients, degradation of food material, defence against natural enemies, and increase insects’ fitness [261,262,263]. Obligate ectosymbionts are stably associated with the insect host, typically localised in specialised host organs. The olive fly ectosymbiont *Candidatus* Erwinia dacicola is a good example [264]. Facultative symbionts do not require a host for survival, may be temporarily associated with the host, are generally horizontally transferred, or acquired from the environment, and may inhabit different organs of the insect (e.g., salivary glands, reproductive organs, etc.) or insect surfaces and play different roles in the insect’s cell cycle. Pseudo-vertical transmission (vertical and horizontal symbionts acquisition) exists for facultative symbionts [260]. Some facultative symbionts are ectosymbionts like *Burkholderia* in the beetle *Lagria villosa* (Fabricius, 1781) (Coleoptera, Tenebrionidae) [265]. The growing scientific interest in the symbiont-insect relationship wishes to extend knowledge on insect biology further and identify new candidates and/or strategies for biological control (biocontrol) of pests.

In this respect, the Red Palm Weevil (*R. ferrugineus*—RPW) has attracted increasing scientific interest in recent decades due to its devastating worldwide invasion of palms, resulting in severe economic issues [266]. Studies on the interaction between RPW and microorganisms mainly follow two research targets. The first search for natural enemies for RPW biological control, and the second identifies ectosymbionts to disrupt their role in the RPW life cycle and insect-plant interaction [267,268,269,270]. Biological control emerges as an alternative to conventional management based mainly on using chemicals, which entails severe concerns for human health, environmental pollution, and selecting resistant insects.

The natural enemies of RPW, here restricted to microorganisms, are bacteria, fungi, and viruses. Only one virus, *Cytoplasmic Polyhedrosis Virus* (CPV), is infectious in all stages of RPW [271]. However, data on CPV for biocontrol of RPW are limited to laboratory tests. As for fungi, *B. bassiana* and *M. anisopliae* are the two entomopathogenic fungi mainly studied for biocontrol of RPW [272]. *B. bassiana* can infect RPW eggs, larvae, and adults and be transmissible among adults. We expect that *B. bassiana* significantly affect the RPW population in infested palms, reducing the number of adults and their reproductive efficiency. Still, Besse [273] suggests that commercial oil formulation of *B. bassiana* has moderate results in the field. *Metarhizium anisopliae* is also a promising biocontrol agent for RPW [272]. Laboratory data confirmed the efficacy as high mortality *M*. *anisopliae*-treated RPW larvae and adults. A recent oil-in-glycerol formulation of *M*. *anisopliae* proposes a possible field application. The formulation is stable over time and can prolong conidial shelf-life compared to unformulated conidia [274]. Potential pathogenic bacteria for RPW mainly belong to the Gram-positive *Bacillus* sp., Gram-negative *Serratia* sp., and *Pseudomonas aeruginosa* [261]. The available data only concern studies conducted under laboratory conditions; *Bacillus thuringiensis*, *Bacillus sphaericus*, *Serratia marcescens*, and *P. aeruginosa* are the primary bacteria employed in bioassays and proposed as candidates for the biological control of RPW. Although promising, the use of bacteria for biocontrol is still debated, particularly about deployment strategies and human health concerns, as some of the proposed bacteria (e.g., *S*. *marcescens* and *P*. *aeruginosa*) have been responsible for human infections [275,276].

RPW-associated bacteria belong to different Taxa depending on geographic areas, palm species, and collection from larvae, pupae, adults, gut, or reproductive apparatus [23,277].

The widely identified Phyla are Proteobacteria, Bacteroides, and Firmicutes. The role played by RPW-associated bacteria is a topic of interest, which could yield valuable data for implementing new control strategies for RPW management. Considering *S*. *marcescens*, a facultative ectosymbiont of RPW, leads to understanding the role of bacterial symbiosis with the weevil [23]. The *S. marcescens* strains associated with RPW produce or not the red pigment prodigiosin and were regularly released during oviposition by females, as demonstrated through in vivo experiments with apples provided as the substrate for oviposition. The same Red-Pigment-Producing *S. marcescens* (RPPS) strains also exist in the reproductive apparatus and gut of dissected adult and virgin RPW and on the internal surface of pupal cases collected from infested palms. Strains of RPPS widely spread along the tissues of infested palms. Extensive studies have reported the antimicrobial activity of prodigiosin [278,279]. This finding is consistent with the antibacterial activity shown by RPPS strains collected from RPW vs. both Gram-positive (*Bacillus* sp., *Paenibacillus* sp., *Lysinibavillus* sp. and *Staphylococcus aureus*) and Gram-negative bacteria (*Escherichia coli*, *Salmonella typhimurium* and *Klebsiella pneumoniae*). In addition to the presence of a red pigment, the study conducted by Scrascia and colleagues [144] showed production from *S. marcescens* symbionts of a possible additional molecule (yet to be identified and characterised) with antibacterial properties. It is noteworthy that *Serratia* from RPW exhibits its antibacterial activity vs. *Bacillus* sp., which is a potential agent for the RPW biocontrol. Some strains of *S*. *marcescens* are known to produce volatile organic compounds with cytotoxic inhibitory activity against pathogenic bacteria, fungi, and nematodes [280].

Moreover, genetic information about encoding enzymes for plant polymer degradation exists in the genome of *S*. *marcescens* [281]. Thus, more than a pathogen of RPW, *S*. *marcescens* could play a role in the RPW life cycle, ranging from protection against natural enemies (due to its antibacterial properties) to metabolic abilities that would influence the interaction between insects and plants. Indeed, both cellulolytic activity and fermentative metabolism (extensively reported for *S. marcescens*) would allow the spread of RPW larvae within the palm tissues and explain the temperature increase detected within infested palms, favouring the insect’s development [282,283].

## 13. Aphrophoridae Froth Niche

The juvenile instars of the superfamily Cercopoidea, known as froghoppers or spittlebugs, live in a surrounding liquid frothy “pond”. Juveniles inject air bubbles into the submerging self-produced fluid. The references [284,285,286,287,288] suggest that froth originates from abdominal gland ducts and liquid faeces.

Tonelli [289] analysed the microbiological composition of the bacteria inhabiting the foam collected from nymphs of *Mahanarva fimbriolata* (Stål, 1854) (Hemiptera, Cercopidae). The molecular analysis of microbial community structures generated OTUs (Operational Taxonomic Units) and found three of the most representative 257 OTUs (relative abundance > 2%) to belong to Alphaproteobacteria (29.9%), Actinobacteria (14.0%), and Chloroacidobacteria (6.3%). Tonelli [289] also replicated the extraction of nucleic acids from the nymph gut and the underlying soil (291 and 288 OTUs). The comparison of the community structures obtained from the three environments showed that the OTUs’ composition of the froth has more in common with the insect’s gut (48 OTUs) than with the underlying soil (24 OTUs). The results exclude that the microbes in the froth originate from the soil. However, they point to gut communities as the primary source of microorganisms.

In the last decade, another family of Cercopoidea has come under the spotlight of the scientific community. Few Aphrophoridae play a primary role in transmitting xylem-inhabiting pathogens. *Xylella fastidiosa* subsp. *pauca* ST53 is the causal agent of OQDS in Italy and entered through an infected host plant or its adult vector [290]. Aphrophoridae species, notably *P. spumarius*, have become the key pests of Mediterranean olive trees [182].

The microbial community of the Juvenile Aphrophoridae Froth (JAF) and its symbiotic benefits remain unknown. Our results rise from the systematic sampling and inoculation in the Petri dishes of field-collected *P. spumarius* froth in sterile test tubes in 2021–2022. We plated the foam in Petri dishes on nutrient agar medium (Thermo Fisher Scientific, Waltham, MA, USA) by soaking sterile stabs.

Observation of the plates enabled the morphological recognition of three repeated colour patterns: violet-yellow, ochre-yellow, and straw-yellow (Figure 3). Firstly, Next-Generation Sequencing (NGS) identified the isolates belonging to these patterns; the sequences obtained made it possible to determine by database comparison (NANOPORE-WIMP) the presence of the genera: *Microbacterium* (Actinobacteria, Actinomycetia), *Pseudomonas* (Proteobacteria, Gammaproteobacteria) and *Agrobacterium* (Proteobacteria, Alphaproteobacteria). The other isolates’ complete identification could enrich this experimental result.

The froth mass forms a barrier to the diffusion of atmospheric O_2_ through the foam. The gaseous exchanges of juveniles occur by extending the abdomen outside the spittle mass. The insect then retracts the tip of its abdomen into the foam mass, producing new air bubbles in which the internal O_2_ pressure is lower than atmospheric pressure [291]. This aspect makes the froth environment even more restrictive to colonisation.

It was necessary to recreate such growth conditions to validate the other microbial characteristics required to survive or grow in the froth environment. Oxoid™ AnaeroJar™ (Thermo Fisher Scientific, Waltham, WA, USA) allowed the plate inoculation of JAF under controlled anaerobic and microaerobic conditions. The varying oxygen availability unveiled additional bacterial isolates currently being identified (Figure 4). We do not exclude the possibility that such organisms are the same as those isolable at regular oxygen rate but with the option of secondary micro-aerobiosis/anaerobiosis.

Spittlebug nymphs inflate air bubbles in a self-secreted and egested liquid containing 99.30–99.75% water and Malpighian protein molecules [292]. The presence of these bacterial genera in the spittlebugs’ froth implies the ability of microbes to utilise the substances in the foam.

Physiochemical functions have already justified the presence of proteins within the Cercopoidea froth matrix. Adequate water surface tension to maintain the froth’s structure is allowed by the presence of the proteins in the foam of *Callitettix versicolor* (Fabricius, 1794) (Hemiptera, Cercopidae) [293,294,295]. However, microorganisms living in peculiar environments such as JAF could exploit such proteins.

Microbacteriaceae (*Microbacterium* spp.) are widespread bacteria. They have demonstrated a marked ability to utilise a wide range of substrates, and isolations have been reported from various matrices: air, water, soil, milk, phyllosphere, and insect gut [296].

*Pseudomonas* spp. and *Agrobacterium* spp. are two ubiquitous genera, often found in soil and the phyllosphere. Contact contamination with these substrates can explain their presence in the JAF. However, a relationship with the insect is not excluded, as in the reported cases of a close host-crop relationship [297,298].

Defining a constancy in the insect class concerning symbiosis with gut microorganisms is difficult. In Hemiptera, it is possible to find cases in which some sap-feeders have little or no gut microbiota but depend on intracellular symbionts for specific nutrients [10]. The primary role of gut bacteria may supply functional components or participate in the digestion and detoxification of host harmful substances [299].

The excretory organs of Insects are the Malpighian tubules that extend into the hemocoel absorbing wastes, such as uric acid, pouring them into the hindgut for disposal. The hindgut manages a combination of nitrogenous and food waste, creating a proper environment for hindgut bacteria, leading to sorting differently from the foregut [300].

Therefore, the foam produced by Aphrophoridae still needs to be explored and defined on a microbial scale. Its chemical composition and the rarefied oxygen make it a suitable bacterial microhabitat. Such an ecological niche may host only well-adapted organisms that could have close relationships between microbes and the host insect. Finally, we cannot exclude that this ectosymbiosis may have evolved into a mutual benefit for both (e.g., protective antimicrobial synthesis) [301].

## 14. *Aureobasidium* spp.: Multitasking Beneficial Microorganism

The yeast-like fungal genus *Aureobasidium* is naturally widespread in the carposphere and phyllosphere of fruits and vegetables. *Aureobasidium* species possess different enzymatic patterns closely related to biotechnological and agricultural uses [302]. These features favour their culture and employment as biocontrol agents against pathogenic fungi and pests [303].

Fungi, especially *Aspergillus* spp., *Cladosporium* spp., *Penicillium* spp., *Rhizopus* spp., and *Aureobasidium* spp., contribute to the pollen composition. These fungi use extracellular enzymes to convert proteins, carbohydrates, and fats into antibiotics, organic acids, and other metabolites [304]. Within the *Aureobasidium* genus, several fast-growth, dimorphic fungi have poly-extremotolerant properties and produce mainly yeast-like cells involved in melanin wall deposits and are therefore called “black yeasts” [303]. The most widespread species is *A. melanogenum*, followed by *A. pullulans*. *Aureobasidium* spp. can synthesise different enzyme patterns, depending on species and strains [303]. *A. melanogenum* produces pullulan, which is involved in resistance to environmental stresses such as UV radiation, high salt concentration, desiccation, strong oxidation, and heat. The strain TN3–1 is a large producer of pullulan and was isolated from natural glucose-rich honey. Furthermore, this strain disclosed high osmotic tolerance due to small vacuoles, trehalose, and glycerol in its cells [305]. From xylose, glucose, and sucrose, *A. melanogenum* biosynthesises liamocin, related to the release of Massoia lactone [303], the last compound being effective against *Fusarium* head-blight [303,306] and with larvicidal action against *Aedes aegypti* L., 1762 (Diptera, Culicidae) [307].

These metabolites can implement sustainable control means, counteracting the pollinator decline due to the depletion of plant biodiversity. Indeed, plant biodiversity depletion significantly reduces pollinators’ nesting habitat and food resource availability [308]. Additionally, these means help prevent resistance to pesticides, such as insecticides, herbicides, and fungicides, that link closely to the target-specificity efficacy of these compounds [309,310].

Among *A. melanogenum*, strain CK-CsC is isolated from honeybee bread and produces cellulase, lipase, amylase, polygalacturonase, xylanase, proteinase, transferases, and mannanase, well-known as food additives [311]. Using this strain as a potential probiotic in the diet of honeybees (*Apis mellifera* L., 1758; Hymenoptera, Apidae) would improve the diet’s nutritional properties and honeybees’ health [311]. In addition, *A. pullulans* displays antifungal activity that controls some fungal pathogens (*Rhizoctonia solani*, *Monilinia* spp., *Neofusicoccum parvum*, and *B. cinerea*) and promotes plant growth.

*A. pullulans* effectiveness is related to non-volatile and volatile organic compounds (VOCs), such as pullulan, degrading enzymes, siderophores, and aureobasidins. However, these VOCs are mainly responsible for the antifungal properties of *A. pullulans*. The formation of pullulan biofilm prevents fungal attachment and colonisation [312]. The antifungal activity of *A. pullulans* was also evaluated on strawberries to control root and crown rot and grey mould caused by *Phytophthora cactorum* and *Botrytis cinerea*, respectively. *B. cinerea* infects strawberry flowers and remains latent until optimal environmental conditions and fruit ripening occur. Therefore, during the blooming stage, applications of chemical fungicide solutions [313] or biocontrol agent (BCA) suspensions are required but adequate covering of the flowers is not guaranteed [314]. Then, the dispersal of BCA by pollinating insects could help reach flower cavities without water use [315]. In this 2-year research, bumblebees (*Bombus pratorum* L., 1761; Hymenoptera, Apidae) used as a carrier were tested in the field, comparing two different dispersion devices and demonstrating their similar dispersal efficacy. The cells of *A. pullulans* spread by the bumblebees during the blooming ranged between 10^3^ and 10^5^ cells per blossom. The efficiency in controlling grey mould ranged between 60–80% [315]. Similar results were obtained by Iqbal [310], evaluating the congeneric species *Bombus terrestris* (L., 1758) (Hymenoptera, Apidae) as a carrier of the same BCA. Entomovectoring methods for dispersing BCA may ensure high precision in reaching flowers at the right time and reduce treatment costs by saving water and reducing the amount of product needed by 80 to 90%. In addition, dry application avoids moisture that could promote fungal infections [310]. Generally, bumblebees are better entomovectors than honeybees at low temperatures due to their higher metabolism in adverse thermal conditions. During the flight, the weight resistance of bumblebees with and without BCA loading was evaluated, which showed no significant differences and validated their loading capacity [310].

Finding new species of *Aureobasidium* spp. and improved culture conditions would make it possible to improve the extraction of natural metabolites with further biotechnological applications. This yeast-like fungus and/or related substances would be facilitated by its easy growth in a bioreactor with a liquid medium, avoiding the problems caused by low oxygen supply and concerns due to complex handling typical of filamentous fungi. *Aureobasidium* spp. derivatives are promising candidates to replace chemical pesticides to safeguard the environment and human and animal health. Furthermore, this eco-friendly One Health approach could encourage organic farming to improve the pollinating insect life cycles and promote biodiversity. Finally, bumblebees and entomovectoring methods could be the new frontier in the dispersion of biocontrol agents, with economic benefits.

## 15. Conclusions

Insects and microorganisms have a long history of interactions, and studying these phenomena provides a multifactorial view to deepen and complement the knowledge already acquired about the actors involved. These interactions act as an evolutionary engine that also makes it possible to overcome environmental stresses such as climate change. Indeed, developing strategies to safeguard the most vulnerable insect species could include the use of symbionts that can enhance their fitness towards a resilient or antifragile ecological response. The insects’ reply consists of hosting microorganisms in their body as endosymbionts (ENS), living in the host’s tissues and often infecting both the insect and the plant, or as ectosymbionts (ECS) living out of the body wall and often out of the cuticle. For example, selected strains of *Aureobasidium* produce probiotics that can improve the health of some pollinators by increasing their fitness to cope with biotic and abiotic stresses in agroecosystems.

Modifications induced by certain microorganisms can improve the environmental conditions favourable to the post-embryonic development of insects, as could be the case with the bacteria inhabiting the Aphrophoridae froth. Furthermore, they may establish antagonistic symbioses between the host plant and the pests, improving their fitness, as *S. marcescens* does with *R. ferrugineus*.

The microbial composition of juvenile aphrophorid foams could play an active role in the survival of naiades to abiotic or biotic agents. Their metabolisms could actively modify surface tension or add antibiotic or repellent effects, the latter case already documented by the topical irritability of palmitic and stearic acids found in the foam.

The JAF represents a hypoxic environment; rarefied oxygen selects microorganisms characterized by microaerophilia or anaerophilia. The scarcity of oxygen also joins dense glandular secretions from Batelli glands and nutrient scarcity because of the xylem sap origin of faeces and insect first food exploitation.

The set of microorganisms specialised to live in such an environment and the interactions between them and the host constitute a micro-niche with its inputs and outputs of energy and matter. Such conditions form annually and persist for a limited time (equal to the post-embryonic development of its host). The origins and ways such microorganisms survive in the absence of JAF and reoccur in the following year remain unknown.

Some plant pathogens can exploit insects to spread in the environment to generate new infectious events. Phytopathogenic endosymbionts such as *X. fastidiosa* only carry out their contagious process in the presence of efficient insect vectors capable of acquiring and bearing them from plant to plant. Other plant pathogens can exploit the dissemination by insects such as ectosymbionts, e.g., *B. oleae* and *Colletotrichum* spp. or *L. botrana* and fungal rots agents.

The microbial communities that compose the microbiome of some insects can improve the fitness and adaptability of some pests, allowing them to do the host shift, e.g., *A. spiniferus* with *A. altissima*.

In addition, some microorganisms can interact negatively with the fitness of the host insect by being used as biocontrol agents. Entomopathogenic microorganisms (fungi, bacteria, or viruses) can reduce the incidence of certain pests (*R. ferrugineus*, *C. sordidus*, *R. palmarum*, *etc.*), reducing agricultural losses and allowing an eco-friendly approach.

The relevant detail consists in the places the microorganisms use to thrive with the insect. Despite the attention to ENS, living in the host’s tissues and often infecting both the insect and the plant, a considerable amount of information suggests that ECS may be a more exploitable guild of guests for the sake of damage management. On the opposite side of the story, advantages may exist in exploring new associations among chosen ectosymbiotic bacteria and insects reared for food or feed, exalting the fitness of the final insect consumer.

Two models of control action depend on the insect-microorganism interaction. ENS are well-specialized multi-host restricted actors. Still, ECS shows many interactions, ranging from the simple microorganism driven by contamination to many morphologically complicated stories, i.e., *Candidatus* Erwinia dacicola or phoronts cascades as in *Rhynchophorus* spp. The RPW reminds the unexpected positive effects of its damage prevention offered by thiophanate-methyl were explained by observing that females contaminate the eggs at laying with a blend of yeast living in the lumina of female genitalia. Remarkably, females also release *Serratia* spp. and Nematoda at the same time.

The case suggests exploring other pest bionomics for similar associations because ECS management and replacement can easily disrupt the proper pest association, lowering the target insect fitness or uncoupling the pest from sensitive habitats.

Entomovectoring for disrupting pest-favourable natural ectosymbiotic interrelationships appears much more feasible, effective, and less impacting on wide-area context than formulates. Furthermore, disruption will act more easily *ex-ante* the pest damage supporting the antifragile intent for a gentle, non-invasive influx on artificial habitats.

In addition, certain VOCs produced by pathogenic invertebrate microorganisms (entomopathogenic or nematophagous fungi) can be used as repellents to manage high economic interest pests (*R. ferrugineus* and *C. sordidus*), generating a ‘green’ biotechnological means of pest control, and reducing food losses.

Therefore, the ectosymbiotic interactions between microorganisms and insects impact the evolutionary history of the actors involved [316]. Interactions can be a tool for the service of humankind to improve its impact on the environment. We must remember that generalist insect-driven microorganisms, much more than specialised insect-transmitted ones, cause the bulk of the damage to humankind. A better understanding of these interaction mechanisms is now possible by basic techniques of microorganism extraction from insect compartments and specialised structures coupled with rapid NGS sequencing of the whole available genome from the collecting site on the insect. Thus, combining techniques and re-discovering effective approaches to observations would enable scientific and technological progress to benefit our common home, the Earth.

## Figures and Tables

**Figure 1 microorganisms-11-00440-f001:**
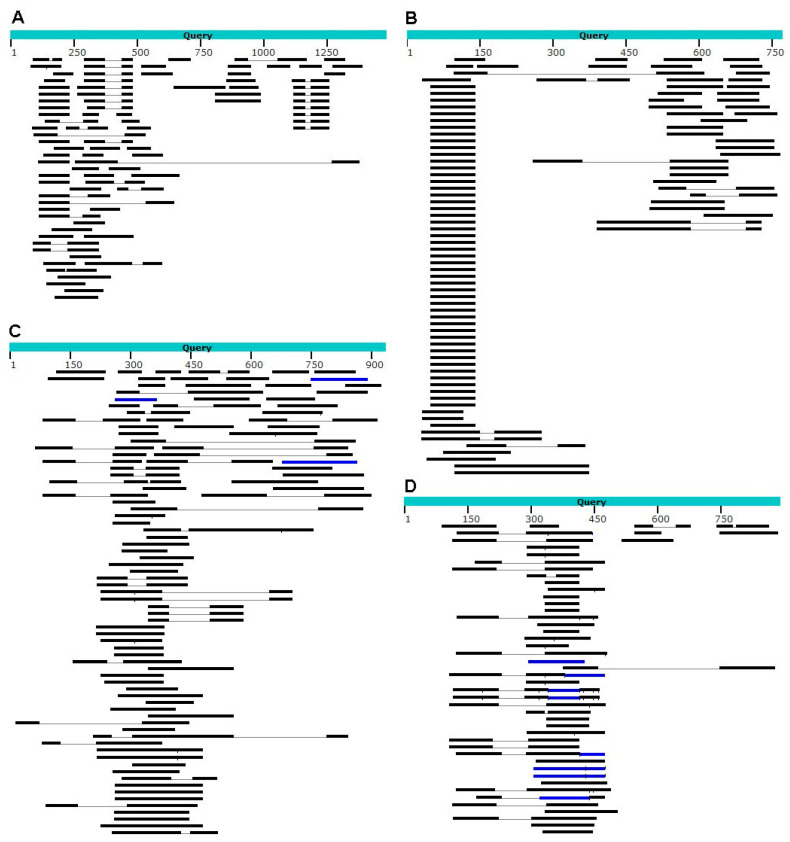
Distribution of best matches produced by TBLASTX on the *P. spumarius* PSPU08 genome using query sequences of nudivirus PIF proteins. Top hits of *N. lugens* nudivirus PIF-1 (**A**) and PIF-2 from nudiviruses of *Drosophila* sp. (**B**), *Macrobrachium rosenbergii* (**C**), and *D. melanogaster* (**D**). NCBI accession numbers of query sequences are KJ566575.1 (**A**), MT496843.1 (**B**), JQ804993.1 (**C**), and NM_169147.2 (**D**). Colours indicate levels of NCBI alignment scores (black: <40; blue: = 40–50).

**Figure 2 microorganisms-11-00440-f002:**
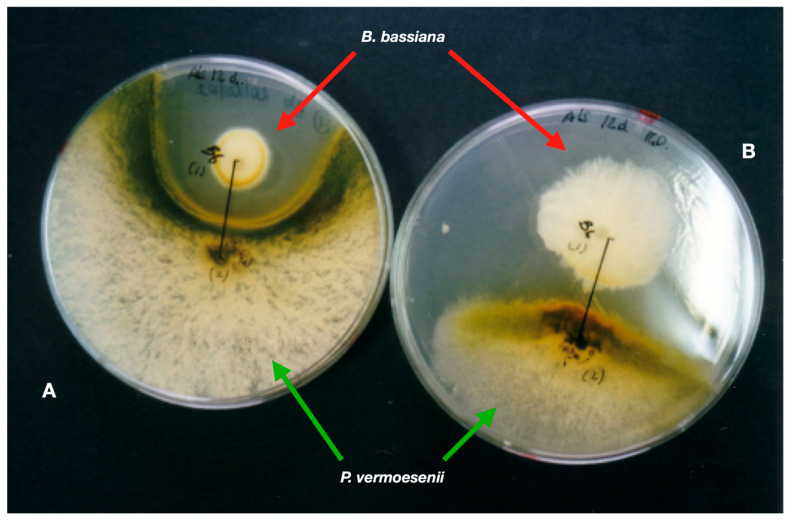
*Beauveria bassiana* (red arrows) inhibits the fungus palm pathogen *Penicillium vermoesenii* (green arrows). (**A**) Both fungi interact directly on the PDA medium. (**B**) The same two fungi on top of a dialysis membrane overlaid onto PDA.

**Figure 3 microorganisms-11-00440-f003:**
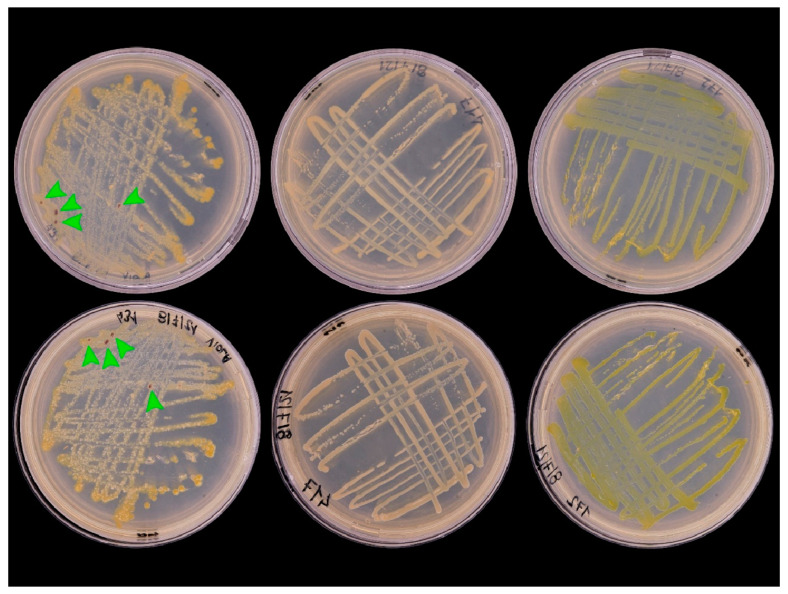
Bacterial isolates in Petri dishes from JAF, identified as genera (from left to right): *Pseudomonas*, *Agrobacterium*, *Microbacterium*; upper and lower Petri sides; green arrows indicate purple spots of the genus *Pseudomonas*.

**Figure 4 microorganisms-11-00440-f004:**
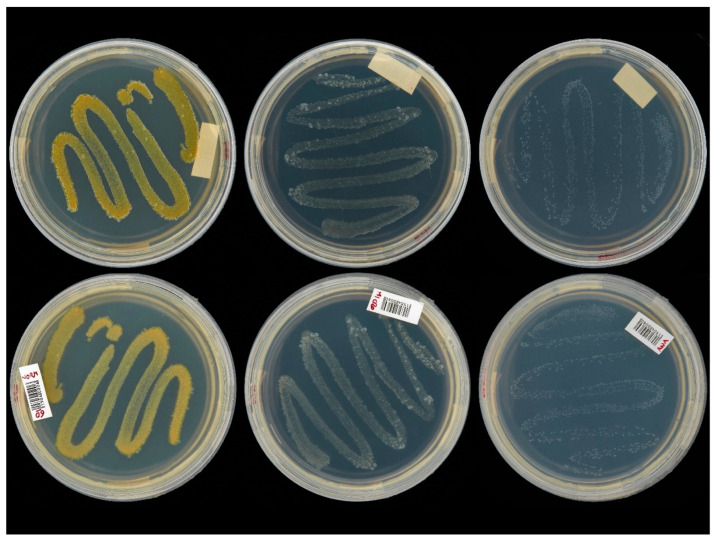
Bacterial isolates obtained by inoculation of unique JAF on nutrient agar incubated under uncontrolled oxygen (**left**), microaerophilic (**middle**), and anaerobic (**right**) conditions, respectively, using Oxoid™ AnaeroJar™; upper and lower face.

**Table 1 microorganisms-11-00440-t001:** Results of PubMed interrogations with query terms combinations ^1^.

Query Terms	*Wolbachia*	*Buchnera*	*Rickettsia*	*Cardinium*	Endosymbiont
Weevil	33	1	34	2	56
*Rhynchophorus*	0	0	1	0	2
*Cosmopolites*	0	0	0	0	0
*Philaenus*	3	0	2	1	2

^1^ Dated 31 August 2022; default parameters.

**Table 2 microorganisms-11-00440-t002:** Top matches ^1^ of translated nudivirus PIF proteins, from different arthropod hosts, on the *P. spumarius* genome ^2^.

Query Protein	Virus	Acc. n.	N. of Matches	Max id. (%)	Lowest E-Value
PIF-1	*Nilaparvata lugens* endogenous nudivirus, isolate Hangzhou	KJ566575.1	87	60.0	0.002
PIF-2	*Drosophila*-associated nudivirus, isolate UA_Kan_16_57	MT496843.1	100	60.0	0.001
PIF-2	*N. lugens* endogenous nudivirus, isolate Hangzhou	KJ566558.1	100	52.0	2 × 10^−4^
PIF-2	*Hyposidra talaca* nucleopolyhedrosis virus, isolate Hyta NPV-ID1	MT642700.1	8	45.7	0.004
PIF-2 (putative)	*Macrobrachium* nudivirus CN-SL2011	JQ804993.1	100	65.0	1 × 10^−4^
PIF-2 (mRNA)	*D. melanogaster* PFTAIRE	NM_169147.2	50	60.0	2 × 10^−6^
PIF-3 (complete cds)	*N. lugens* endogenous nudivirus, isolate Hangzhou	KJ566581.1	67	81.8	0.001
PIF-3 (putative, mRNA)	*Cotesia congregata*	FM201563.4	100	71.4	6 × 10^−5^
PIF-4	*N. lugens* endogenous nudivirus, isolate Hangzhou	KJ566551.1	26	60.0	0.001

^1^ Based on TBLASTX analyses of open access data available at https://www.ncbi.nlm.nih.gov/genome/7381 (accessed on 1 September 2022). ^2^ GenBank assembly GCA_018207615.1 (PRJNA602656) produced by Biello [60].

**Table 3 microorganisms-11-00440-t003:** Several records of products for arthropod control in Brazil.

Product Records	2022	2021	2020	2019	2018
Insecticide	705	71	53	50	51
Microbiological insecticide	238	44	42	18	23
Biological Control Agent	69	6	3	6	5
Microbiological fungicide	66	19	15	6	8
Pheromone	46	2	1	0	3
Microbiological nematicide	46	6	12	6	1
Microbiological acaricide	42	12	10	6	2
Microbiological bactericide	5	0	0	0	0

Adapted source [138].

## Data Availability

The authors did not create new datasets. The data are available in the references cited in the text.

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
