# Peer review of "“Ectomosphere”: Insects and Microorganism Interactions"

_microorganisms, 2023, doi:10.3390/microorganisms11020440_

Round 1

Reviewer 1 Report

The Manuscript ID: microorganisms-2166497 reviews many aspects of insects and microorganism interactions. It addresses the interaction between insects and their ectosymbiont microorganisms for environmentally safe plant production and protection. However, as it deals with the subject in broad sense, I'm not convinced it's conveying much new information in any of these areas. Such interactions have been around and its usefulness has been well explored. The authors review much of already documented information and apply it to some samples that reflect their presented figures and tables. The general conclusions drawn from this work do not appear to be especially novel or enlightening, and the specific conclusions regarding the particular sampling with figures under consideration seem very limited.

Thus, I'm not convinced that much new insight is provided by this paper. It does provide an interesting review and some useful sample figures but seems more appropriate as the background information one would use to develop perhaps a class project for students.

In its current format, more insights are needed for its acceptance.

Therefore, I would suggest resubmitting after major revision.

Reviewer 2 Report

This review paper describes the different interactions between insects and their ectosymbiont microorganisms. The whole manuscript is very informative. The authors provide detailed information for each case. The main problem of this review manuscript needs more organization of all the information. The reader will be lost when reading this manuscript. For this reason, this draft needs much work before further submission.

Main comments:

(1). There are 12 parts, and each piece includes several subtitles. I couldn’t understand the order of these 12 parts and their relationships. 

(2). The authors introduce the individual case starting from part 2. However, before introducing each case, I don’t think the authors have explained enough basic information, such as terminology, mechanisms, and study development history. 

(3). Instead of listing individual cases as one part, I would suggest that the authors think about an outline of this manuscript and try to categorize these cases based on their commons and relationships.

(4). It is nice to read each case with detailed information. However, too much information dilutes the focus. I suggest introducing each case briefly.

(5). The authors are expertized in this topic and familiar with every terminology. Please explain every terminology when it is first shown in this manuscript.

(6) take 2.1 for an example. There are six paragraphs under 2.1. Please summarize your scientific language into two or three paragraphs. Please revise other parts throughout your manuscript. 

Other minor comments:

Line 26. Whit? With?

Line 28, since this is abstract, it should be conclusive. I am not sure it is a good idea to point out many examples here, such as “Serratia marcescens does with Rhynchophorus ferrugineus.”

Line 51. Since IPM is first shown here, please provide the full name.

Line 78, if you want to list an example, please provide one sentence about it. Otherwise, you could only put the citation here. Phrases such as “ e.g., Aphididae and Buchnera aphidicola” don’t help readers to understand your topic.

The same problem is in lines 83-84. You could put the citations here. The readers will check the references if they are interested.

Round 2

Reviewer 1 Report

Accept publication

Reviewer 2 Report

Thanks for the revision. The authors try to add more conclusive language in the introduction and conclusion to help readers understand the organization of the whole manuscript well. The manuscript is informative, with multiple examples of interactions between insects and their ectosymbiont microorganisms.

Based on the authors’ reply, they might think this was a topic review, so they present the terms and related knowledge for expert readers. It is good to provide a summarized review for expert readers in your field. However, this publication is supposed to provide information and indication in multiple areas, including insects, bacteria, fungi, viruses, plants, agriculture, biocontrol, IPM, etc. I think that is wonderful for a review paper. So it is supposed to be readable for readers from different fields.

Main comment:

Part 5. Interactions between entomopathogenic fungi and pests & part 6. Multitrophic interactions of entomopathogenic fungi, crops, and insects

Is it reasonable to combine these two parts? Or it will be good to change the subtitle. The current subtitles are conclusive and general, which might equal your paper title. However, your context in these two parts is not conclusive. You still emphasize a specific example.